# Synthesis and Functional Characterization of Co_x_Fe_3−x_O_4_-BaTiO_3_ Magnetoelectric Nanocomposites for Biomedical Applications

**DOI:** 10.3390/nano13050811

**Published:** 2023-02-22

**Authors:** Timur R. Nizamov, Abdulkarim A. Amirov, Tatiana O. Kuznetsova, Irina V. Dorofievich, Igor G. Bordyuzhin, Dmitry G. Zhukov, Anna V. Ivanova, Anna N. Gabashvili, Nataliya Yu. Tabachkova, Alexander A. Tepanov, Igor V. Shchetinin, Maxim A. Abakumov, Alexander G. Savchenko, Alexander G. Majouga

**Affiliations:** 1Department of Physical Materials Science, National University of Science and Technology “MISiS”, 119049 Moscow, Russia; 2Amirkhanov Institute of Physics of Dagestan Federal Research Center, Russian Academy of Sciences, 367003 Makhachkala, Russia; 3Department of Materials Science of Semiconductors and Dielectrics, National University of Science and Technology “MISiS”, 119049 Moscow, Russia; 4Chemistry Department, Lomonosov Moscow State University, 119991 Moscow, Russia; 5Department of Medical Nanobiotechnology, N.I. Pirogov Russian National Research Medical University, 117997 Moscow, Russia; 6Mendeleev University of Chemical Technology of Russia, 125047 Moscow, Russia

**Keywords:** multiferroics, magnetoelectric nanoparticles, magnetoelectric nanocomposites, magnetoelectric effect, iron oxide, cobalt ferrite, barium titanate, cytotoxicity, thermal decomposition, solvothermal synthesis

## Abstract

Nowadays, magnetoelectric nanomaterials are on their way to finding wide applications in biomedicine for various cancer and neurological disease treatment, which is mainly restricted by their relatively high toxicity and complex synthesis. This study for the first time reports novel magnetoelectric nanocomposites of Co_x_Fe_3−x_O_4_-BaTiO_3_ series with tuned magnetic phase structures, which were synthesized via a two-step chemical approach in polyol media. The magnetic Co_x_Fe_3−x_O_4_ phases with x = 0.0, 0.5, and 1.0 were obtained by thermal decomposition in triethylene glycol media. The magnetoelectric nanocomposites were synthesized by the decomposition of barium titanate precursors in the presence of a magnetic phase under solvothermal conditions and subsequent annealing at 700 °C. X-ray diffraction revealed the presence of both spinel and perovskite phases after annealing with average crystallite sizes in the range of 9.0–14.5 nm. Transmission electron microscopy data showed two-phase composite nanostructures consisting of ferrites and barium titanate. The presence of interfacial connections between magnetic and ferroelectric phases was confirmed by high-resolution transmission electron microscopy. Magnetization data showed expected ferrimagnetic behavior and σ_s_ decrease after the nanocomposite formation. Magnetoelectric coefficient measurements after the annealing showed non-linear change with a maximum of 89 mV/cm*Oe with x = 0.5, 74 mV/cm*Oe with x = 0, and a minimum of 50 mV/cm*Oe with x = 0.0 core composition, that corresponds with the coercive force of the nanocomposites: 240 Oe, 89 Oe and 36 Oe, respectively. The obtained nanocomposites show low toxicity in the whole studied concentration range of 25–400 μg/mL on CT-26 cancer cells. The synthesized nanocomposites show low cytotoxicity and high magnetoelectric effects, therefore they can find wide applications in biomedicine.

## 1. Introduction

During the last decade, magnetoelectric (ME) materials have attracted significant interest for application in various branches of science and technology, such as magnetic-field sensors, spintronics, energy harvesters, and biomedicine [1,2,3,4,5]. This is due to their unique coupling of magnetic and electric properties. This coupling causes a specific phenomenon called the ME effect—a conversion of electric to magnetic energy and vice versa. There are two types of ME materials by structure: single-phase and two-phase materials. The first has magnetic and electric orders simultaneously in one phase, such as BiFeO_3_ [6] and YMnO_3_ [7]. The second has two separated phases consisting of ferro or ferrimagnetic and ferroelectric materials with magnetic and electric orders, respectively [8]. Usually, ferrites are used as a magnetic phase, such as CoFe_2_O_4_ and perovskites as a ferroelectric similar to BaTiO_3_. Two-phase ME materials, known as ME composites, attract more attention due to a significantly higher ME effect compared to single-phase materials because magnetic and electric orders must coexist in an asymmetric structure in single-phase materials. The ME effect itself is a result of coupling between the magnetostrictive effect, which is the property to change shape during the process of magnetization in the magnetic phase; and the ferroelectric effect, which is the property to polarize under mechanical deformation in the ferroelectric phase [9]. Therefore, the two key parameters affect the coupling and ME effect, respectively—interface connectivity, inner magnetic, and ferroelectric properties of subsequent phases. 

In connection with the biomedical applications of ME composites, a novel concept, based on the idea of using both magnetic and electric fields to control drug delivery and release processes, was presented in the publications [10,11,12,13]. In this approach, ME composites were referred to as ME nanoparticles (MENs) having a core-shell structure and used as functionalized nanomaterials. MENs are nanocomposites with a core-shell structure consisting of a magnetic core and a ferroelectric shell. The prospects of MENs, from synthesis to application, were described in the reviews [14,15,16].

The core-shell structure is expected to have a better arrangement of magnetic and ferroelectric phases because the higher interface contact leads to a higher ME effect [3]. Thus, every single nanoparticle of the magnetic phase should be covered with a ferroelectric shell. Currently, novel methods of core-shell ME nanocomposites (MENC) are being actively developed, especially on the basis of the CoFe_2_O_4_ core and BaTiO_3_ shell [17,18,19,20]. MENC, with a core-shell structure, attracts particular interest for application in biomedicine. This is not only due to the expected high ME effect but also because of the possibility to overcome specific size restrictions for biomedical applications, as core-shell particles, which are dozens of nanometers in size, enable ME composites to interact with cells, and they selectively affect cellular receptors and membranes on a nanoscale. MENC applications in biomedicine have been well shown. They have been used for ME effect-triggered drug release, especially for cancer treatment in vitro [10,17,20] and in vivo [11,12,21]. Moreover, they are used in other biomedical fields as well, such as the brain [22] and magnetic particle mapping [23]. Unfortunately, the majority of the ME materials, such as CoFe_2_O_4_-BaTiO_3_, consist of rather toxic elements: cobalt and barium. While iron [24,25,26] and titanium-based [27,28] materials are considered to be biocompatible and relatively non-toxic and are therefore widely used in biomedicine, the application of materials made of cobalt [29,30] and barium [31] is restricted due to the lack of biocompatibility. The decrease of ME material cytotoxicity or the replacement of ME materials with less toxic ones is a current challenge in the field.

In all the cases mentioned above, cobalt ferrite has been used as a magnetic core due to its high magnetostriction, which correlates with the magnetocrystalline anisotropy of the magnetic atoms in the lattice. For Co^2+^_,_ the value is 1.8–3 × 10^5^ J/m^3^ at 300 K [32], which is more than 10 times higher than for Fe^2+^ [33]. The anisotropy also affects the coercivity of magnetic materials. While in general the rule: “the higher number of atoms with high magnetocrystalline anisotropy, the higher coercivity will be” works, in some cases for non-stoichiometric cobalt ferrites higher coercivity was shown in compounds with a smaller amount of Co^2+^ compared to stoichiometric compounds [34,35]. Moreover, the effect of magnetic core stoichiometry on ME effect has not been studied yet.

However, the methods of ME composite synthesis for biomedical applications mentioned above are largely based on citrate synthesis; magnetic nanoparticles are usually synthesized by coprecipitation of titanium and barium citrates with subsequent evaporation of the solvent. The final mixture of citrate complexes and magnetic phase is annealed, resulting in the formation of a core-shell structure [12]. Unfortunately, due to the usual mismatch of their lattices, the formation of direct interphase contact between the phases is a serious challenge [8,36]. Moreover, it is still unclear how during solvent evaporation the colloidal magnetic nanoparticles are electrostatically stabilized by citrate-ions, and Ba and Ti citrate complexes can coprecipitate together and form a homogeneous mixture. Usually, colloids stabilized this way are sensitive to ionic strength change which dramatically increases during evaporation and subsequently leads to the preliminary aggregation of the magnetic nanoparticles, and then to the decrease in interphase connections. At the same time approaches based on the synthesis of magnetic nanoparticles in polar organic media have also been described, especially in glycol media [34,35,37]. This type of media is suitable for the synthesis of BaTiO_3_ nanoparticles as well [38,39,40]. Therefore, we decided to obtain the primary structure, which is close to the core-shell, via the covalent binding of Ba and Ti precursors onto the magnetic core surface in solvothermal conditions. The use of the magnetic phase with no strong stabilizer in a mild glycol medium compared to water might ease the process of formation of interphase connections. The final BaTiO_3_ phase and surface connections between phases were obtained by annealing. Moreover, MENC with three different types of Co_x_Fe_3−x_O_4_ magnetic core with x = 0.0; 0.5; 1.0 were synthesized to study the effect of magnetic phase on the ME effect. 

## 2. Experimental Section

### 2.1. Chemicals

Iron (III) acetylacetonate (Fe(acac)_3_; Sigma-Aldrich, St. Louis, MO, USA), cobalt (II) acetylacetonate (Co(acac)_2_; Sigma-Aldrich, St. Louis, MO, USA), triethylene glycol, 99% (TEG; Sigma-Aldrich, St. Louis, MO, USA), titanium (IV) isopropoxide (97%, Sigma-Aldrich, St. Louis, MO, USA), barium hydroxide octahydrate (II) (Ba(OH)_2_·8H_2_O; Sigma-Aldrich, St. Louis, MO, USA), 1-octadecene, ≥95% (Sigma Aldrich, St. Louis, MO, USA), oleic acid, ≥99% (Roth, Karlsruhe, Germany), ethanol, ≥95% (Sigma Aldrich, St. Louis, MO, USA), MTS (Promega, Madison, WI, USA), NaOH, 99.5% (Sigma Aldrich, St. Louis, MO, USA), polyvinylpyrrolidone, 10 kDa (PVP, Sigma Aldrich, St. Louis, MO, USA), Pluronic F127 (Sigma-Aldrich, St. Louis, MO, USA), and deionized water.

### 2.2. Synthesis of the Samples

#### 2.2.1. Synthesis of Co_x_Fe_3−x_O_4_ Phase

The magnetic phase was synthesized according to a slightly modified version of the protocol [35]. Ferrite seeds of Co_x_Fe_3−x_O_4,_ where x = 0.0 (x = 0.0-0); 0.5 (x = 0.5-0); 1.0 (x = 1.0-0), were synthesized by thermal decomposition of x × 4 mmol Co(acac)_2_ and (3 − x) × 4 mmol Fe(acac)_3_ in 120 mL TEG medium at 265 °C for 1 h under argon blanketing and cooled to room temperature.

The final magnetic nanoparticles (MNP) of Co_x_Fe_3−x_O_4_ were synthesized via dropwise injection of x × 12 mmol Co(acac)_2_ and (3 − x) × 12 mmol Fe(acac)_3_ in 240 mL of TEG into 60 mL of seeds solution at 225 °C under argon blanketing for 1.5 h. Then, the reaction medium was kept at 225 °C for an extra 30 min and cooled down to room temperature. Finally, three types of MNP with Fe_3_O_4_ (x = 0.0-1), Co_0.5_Fe_2.5_O_4_ (x = 0.5-1) and CoFe_2_O_4_ (x = 1.0-1) structure were obtained.

#### 2.2.2. Synthesis of the Barium Titanate Phase

Co_x_Fe_3−x_O_4_-BaTiO_3_ composites with x = 0.0 (x = 0.0-BT); 0.5 (x = 0.5-BT); 1.0 (x = 1.0-BT) were synthesized using a solvothermal approach based on the published protocol with significant modifications [39]. Sixty mL of the MNP sol was mixed with 6 mL of 10% PVP solution in ethanol under stirring. Then the sol was mixed with 120 mL of ethyl acetate and MNP were collected by magnetic decantation. The collected precipitate was redissolved in a mixture of 2.56 mL water and 35 mL TEG and ultrasonicated until a homogeneous solution (MNP-PVP) was obtained. Then, 1.86 mL of titanium (IV) isopropoxide was injected into the MNP-PVP under intensive stirring followed by the addition of 2.38 g of Ba(OH)_2_·8H_2_O in 12 mL of TEG. Solvothermal synthesis was conducted at 200 °C in a 75 mL glass autoclave for 4 h in an oil bath followed by cooling to room temperature. The obtained sol was diluted three times with ethyl acetate and the sample was collected by magnetic decantation for annealing. 

The samples were annealed at 700 °C for 5 h. The annealed samples were dispersed in a 1-octadecene solution with oleic acid (mass ratio sample: oleic acid: octadecene—1:1:8) in a specialized 80-milliliter steel container using 10 balls (d = 10 mm) and 50 balls (d = 5 mm) of tungsten carbide. The mixture was ground in a planetary ball mill (Fritsch pulverisette, Idar-Oberstein, Germany) in the air at 400 RPM for 12 h with a 5 min pause for every hour. The grounded mixture was washed with isopropanol twice, and the magnetic phase was collected by magnetic decantation and redispersed in toluene. 

The grounded samples were hydrophilized through magnetic stirring of 2 mg/mL samples solution in toluene with 2 mM Pluronic F127 solution in water according to the previously published protocol [41]. Finally, the samples were collected by centrifugation and redispersed in distilled water. 

### 2.3. Characterization

The nanoparticles’ morphology and microstructure were imaged via a JEOL JEM-1400 transmission electron microscope (JEOL, Tokyo, Japan) with an accelerating voltage of 120 kV and a JEOL JEM-2100 transmission electron microscope (JEOL; Japan) with an accelerating voltage of 200 kV. Structural phase analysis was carried out on a Rigaku Ultima IV diffractometer (Rigaku, Tokyo, Japan) using CoKα radiation and a graphite monochromator. Static magnetic properties were measured in magnetic fields ranging from −18 to 18 kOe at 300 K on a Quantum Design Physical Property Measurement System (Quantum Design, San Diego, CA, USA). Elemental analysis and concentration measurements were conducted using a 4200 MP–AES machine (Agilent Technologies, Santa Clara, CA, USA). Elemental composition was also studied via SEM/EDX analysis using a scanning electron microscope JSM-6480LV (JEOL, Tokyo, Japan). The hydrodynamic size of the samples was measured by dispersing 50 μL of the as-synthesized colloid sols in 2 mL ethanol or water using a Zetasizer Nano ZS analyzer (Malvern Panalytical, Malvern, UK).

The ME properties were estimated using a dynamic method described elsewhere [42,43]. For this purpose, a custom-designed setup in which modulated AC $H_{AC}$ and biased DC $H_{DC}$ magnetic fields were applied in order to measure the ME voltage changes ΔV of the sample. To detect the ME output voltage, a lock-in amplifier (Model SR830, Stanford Research, Sunnyvale, CA, USA) was used. The ME coefficient $\alpha_{ME}$ was defined using the following equation:$$\alpha_{ME}=\frac{\Delta V}{b\;\Delta H}$$
where ΔV is the amplitude of the induced ME voltage, *b* is the thickness of the sample, and ΔH is the amplitude of the AC-modulated magnetic field $H_{AC}$. The DC bias magnetic field was applied up to 0.5 T. The ME measurements were carried out on a resonance frequency of about 65 kHz and the amplitude of the modulated AC magnetic field was chosen in the range of 1–10 Oe. For the ME studies, grains of samples were ground, mixed, and pressed into disk shape pellets with a thickness of about 0.5 mm and a diameter of 8 mm under a pressure of 100 MPa. All samples were pressed using the same conditions and the presence of the micro-cracks were not observed. Electric contacts were made by coating both surfaces of the disks with special conductive silver adhesive (DUPONT™) as for thin disc capacitors. AC and DC magnetic fields were applied parallel to the sample plane (perpendicular to the poling direction). All measurements were carried out at room temperature.

### 2.4. Cell Line

Murine syngeneic cell-lines of colon-rectum (CT-26) were purchased from ATCC (CRL-2639™). The cell line was cultured in RPMI 1640 media (Gibco, Waltham, MA, USA) supplemented with antibiotics (100 U/mL penicillin, 100 mg/mL streptomycin, Gibco), GlutaMax Supplement (2 mM, Gibco), and 10% fetal bovine serum (HyClone, Cytiva, Washington, DC, USA). The cells were cultured under standard conditions (37 °C and 5% CO_2_) in T-25 cultural flasks (Corning, New York, NY, USA). Upon reaching a high confluence, the cells were subcultured at a ratio of 1:10 according to the standard trypsinization method.

### 2.5. MTS-Assay

Cytotoxicity tests for the samples of x = 0.0-BT, x = 0.5-BT, and x = 1.0-BT in CT-26 cells were performed via CellTiter 96 AQueous One Solution Cell Proliferation Assay (Promega, Madison, WI, USA) according to the manufacturer’s protocol. CT-26 cells were seeded at 12 × 10^3^ cells/well in a 96-well culture plate with 100 μL of medium per well. After 24 h of incubation, the samples of x = 0.0-BT, x = 0.5-BT, and x = 1.0-BT at different concentrations (25 μg/mL, 50 μg/mL, 100 μg/mL, 200 μg/mL, 400 μg/mL) were added to the cells. After another 24 h incubation, the cells were washed with PBS, and a fresh growth medium with MTS reagent was added to each well. Non-treated cells by the samples of x = 0.0-BT, x = 0.5-BT, and x = 1.0-BT were used as a positive control. The cells were incubated in MTS reagent for 4 h at 37 °C and 5% CO_2_ in a humid atmosphere. The assay was conducted in three replicates. Optical density was measured using a Multiscan GO plate reader (Thermo Scientific, Waltham, MA, USA), λ = 490 nm.

Cell viability was calculated as:$$\mathrm{Cell}\;\mathrm{viability}\;\left(\%\right){=(A}_{s}{\;-\;A}_{c})/{(A}_{c}{\;-\;A}_{b})\;\times \;100$$
where A_s_ is the optical density of the sample wells, A_b_ is the optical density of the blank wells, and A_c_ is the optical density of the positive control wells.

## 3. Results and Discussions

### 3.1. Magnetic Phase Synthesis

The synthesis of the magnetic phase was carried out according to the previously published two-step approach [35], where the processes of MNP nucleation and growth are separated for more precise control over nanoparticle size. Three types of Co_x_Fe_3−x_O_4_ seeds: x = 0.0-0; x = 0.5-0; x = 1.0-0, were synthesized in a TEG medium by thermal decomposition of Co(acac)_2_ and Fe(acac)_3_. The average size, as determined by transmission electron microscopy (TEM), was 6.6 ± 1.4 nm, 6.4 ± 1.1 nm, and 5.6 ± 1.0 (Figure 1A–C, respectively). According to dynamic light scattering (DLS) measurements, the average hydrodynamic sizes were 11.8 ± 3.3; 10.7 ± 3.6, and 7.8 ± 2.2 nm (Figure 1D). Initial seeds were colloidally stable and showed no evidence of aggregation. This process was conducted at 265 °C under argon blanketing to avoid any side oxidation reactions.

The obtained black seed solutions, after cooling to room temperature, were used for the second stage of MNP synthesis. The seed solutions were heated to 225 °C, and then the solutions of Fe(acac)_3_ and Co(acac)_2_ were injected dropwise into the seed media. The Co to Fe stoichiometry was kept the same as it was for the initial seeds. Finally, the series of Co_x_Fe_3−x_O_4_ nanoparticles x = 0.0-1, x = 0.5-1, and x = 1.0-1, with average sizes of 15.2 ± 1.9 nm, 13.9 ± 1.6 nm, and 17.1 ± 2.2 nm (Figure 2A–C), respectively, were obtained. The DLS measurements correspond with the TEM data: 15.8 ± 5.8 nm, 21.1 ± 10.6 nm, and 21.1 ± 6.5 nm for x = 0.0-1, x = 0.5-1, and x = 1.0-1, respectively (Figure 2D). The obtained series of MNP show monomodal distribution, no aggregation, and precipitation processes were observed.

Then, the synthesized series of MNP x = 0.0-1, x = 0.5-1, and x = 1.0-1 were studied by X-ray diffraction (XRD) and vibrating sample magnetometry (VSM) measurement. According to structure studies, all samples have a spinel structure (Space group Fd3m) specific for magnetite (PDF Card No.: 01-074-1909) and cobalt ferrite (PDF Card No.: 01-078-4451) as well as other ferrites with the structure: Me(II)Fe(III)_2_O_4_ (Figure 3A, Table 1). The crystallite size decreases with the increase of cobalt content from 12.4 ± 1.2 nm for x = 0.0-1 to 9.1 ± 0.9 nm for x = 0.5-1, and 7.4 ± 0.7 nm for x = 1.0-1. If we compare this parameter with the average size by TEM, the crystallite size is much lower than the TEM size for x = 0.5-1 and especially for the x = 1.0-1. This means that these MNP have a polygranular “nanoclustered” structure [44,45]. The magnetic measurement revealed that all samples are ferrimagnetic materials. The coercive force H_c_ contradictively correlates with Co content: it increases from 49 ± 2.5 Oe for the x = 0.0-1 to 200 ± 10 Oe and the x = 0.5-1, and decreases again to 49 ± 2.5 Oe for the x = 1.0-1) (Figure 3B, Table 2). This tendency was discovered in our previous publication [35]. It can be explained by the smallest crystalline size of 7.4 ± 0.7 nm for x = 1.0-1 in the series, which is a bit more than the superparamagnetic limit for cobalt ferrite, which is approximately 7 nm [46]. It caused a significant decrease in coercive force for this sample, which corresponds with the published data [47]. The polygranular structure might cause an increase in coercive force [48], but in this case, the effect is not significant, probably, due to the low average size of the granules which make up the nanoparticles. The saturation magnetization (σ_s_) and remanence magnetization are significantly high for such size of nanoparticles: 70.9 ± 0.7 A•m^2^/kg, and 5.14 ± 0.05 A•m^2^/kg for the x = 0.0-1; 70.5 ± 0.7 A•m^2^/kg, and 7.66 ± 0.08 A•m^2^/kg for the x = 0.5-1; 66.3 ± 0.7 A•m^2^/kg, and 6.65 ± 0.08 A•m^2^/kg for the x = 1.0-1. The σ_s_ decrease with the increase of Co content, which is reasonable, because the magnetic moment of Co is less than Fe.

The elemental analysis of the ferrite MNP was conducted using the MS AES method. The x = 0.5-1 and x = 1.0-1 samples have a lower amount of Co compared to the loaded ratio of Co to Fe precursors (Table 3). The actual empirical formulas for x = 0.5-1 and x = 1.0-1 are Co_0.42_Fe_2.58_O_4_ and Co_0.93_Fe_2.07_O_4_, respectively. This may be due to the reduction of Fe(III) to Fe(II) by the glycol media, which competes with Co(II) during MNP growth [35,49]. 

All the synthesized samples x = 0.0-1, x = 0.5-1, and x = 1.0-1 were further used for the synthesis of the Co_x_Fe_3−x_O_4_-BaTiO_3_ series.

### 3.2. Investigation of Shell Formation Conditions

As part of this research, a synthesis of final ME composites was carried out according to a modified solvothermal procedure for obtaining a crystalline shell of barium titanate [39]. This technique is based on the hydrolysis of titanium (IV) isopropoxide in polyol media in the presence of Ba^2+^ ions:Ti(OiPr)_4_ + 4H_2_O + Ba^2+^ + 2(OH)^−^ → Ti(OH)_4_ + Ba^2+^ + 4iPrOH → BaTiO_3_ +2H_2_O

Based on the study mentioned, a synthesis of the series Co_x_Fe_3−x_O_4_-BaTiO_3_ in variable conditions, where x = 0.0, was carried out. Therefore, for all the further syntheses x = 0.0-1 samples were used. 

The first series of samples were obtained by the solvothermal synthesis in polyols with the addition of 0%, 5%, and 15% of water by dissolving barium hydroxide in a TEG and the addition of a subsequent amount of water (x = 0.0-BT0%, x = 0.0-BT5%, x = 0.0-BT15%, respectively). All syntheses were carried out using a PVP stabilizer. PVP is a common polymer with low toxicity, which is actively used to stabilize nanoparticles including ferrites. It consists of monomeric units of N-vinylpyrrolidone, which is a lactam (cyclic amide) with lone electron pairs on the nitrogen and oxygen atoms, which, on the one hand, are capable of coordinating with the surface of the magnetic core, and, on the other hand, play the role of a matrix for subsequent formation of the barium titanate shell. The use of this polymer made it possible to significantly increase the stability of the obtained nanocomposites at the stage of solvothermal synthesis. The samples turned out to be relatively polydisperse, but a tendency was revealed by DLS and TEM that the higher the amount of water, the higher the hydrodynamic sizes of nanoparticles turn out to be (Figure 4A,D). According to TEM studies, when the samples were obtained with no water addition, the MNP particles almost completely lacked the barium titanate shell cover. This is because the titanium precursor does not hydrolyze properly when no water is added. The hydrodynamic size of the sample increases to 43 nm compared with the initial MNP, having an average size of 15 increased by 5% by volume nm. When extra water is added in the amount of 5% of the solution volume, the hydrodynamic size of the sample reaches 58 nm and significant shell formation is observed by the TEM study (Figure 4B,D). The extra added water eases the hydrolysis of the titanium precursor and the subsequent formation of barium titanate. The addition of extra water in the amount of 15% of the solution volume leads to the aggregation of nanoparticles with a final hydrodynamic size of ~400 nm, and massive aggregates are observed by TEM (Figure 4C,D). It was concluded that the best water amount for synthesis is 5% because the smaller amount will not lead to a ferrite-barium titanate interfacial connection, while the higher amount causes total aggregation, which can lead to the inhomogeneity of the final MENC. 

The obtained sample (x = 0.0-BT5%) with 5% extra water volume added was further studied by XRD and Energy-dispersive X-ray spectroscopy (EDX) elemental analysis (Figure 5). The sample consisted of barium titanate, iron oxide, and barium carbonate. The last phase was present due to the initial excess of barium in relation to titanium (molar 1.2 to 1.0) and probably not all titanium crystallized into the barium titanate phase and some amorphous titania was present. According to the EDX measurements, the sample contained the elements: Ba, Fe, Ti, and O at a ratio close to what was initially calculated for the solvothermal synthesis, and no components were missed. 

The initial mass ratio of the magnetic phase to the ferroelectric phase was 1 to 2.5. It was chosen because the magnetic phase should be surrounded by barium titanate to gain a higher ME effect, otherwise the presence of interfacial magnetic connections leads to an increase in the electric conductivity of the whole material (ferrites are semiconductors) and a decrease in ME effect. On the other hand, the larger amount of ferroelectric phase leads to less interfacial connections between magnetic and ferroelectric phases and therefore the lower ME effect. Thus, maintaining the optimal ratio of phases is necessary. For this reason, extra samples (x = 0.0-BT 1–4) with 1 to 4 magnetic to ferroelectric phases were synthesized and studied by TEM, XRD, and DLS analyses. According to TEM, the samples were crystalline, and a lack of an amorphous phase was observed (Figure 6A). The nanoparticles formed huge aggregates and rather large spherical nanoparticles of what was probably barium titanate were seen. No core-shell-like structures were observed. The XRD study revealed that the sample was highly crystalline and consisted of barium titanate, iron oxide, and barium carbonate (Figure 6B). The hydrodynamic size of the sample is very large, measuring more than 370 nm, and the colloid is polydisperse (Figure 6C). It could be due to the higher concentration (almost two times higher) of barium hydroxide in the reaction medium in comparison with the standard synthesis. The higher concentration of OH eases the crystallization of barium titanate and its rapid uncontrolled growth. Therefore, it was decided to synthesize the subsequent series of samples via hydrothermal synthesis at a 1 to 2.5 magnetic to ferroelectric phase mass ratio.

### 3.3. Synthesis of MENC Series

The MENC series of x = 0.0-BTO, x = 0.5-BTO, x = 1.0-BTO was synthesized according to the protocol given in the section Materials and Methods. According to the TEM studies the magnetic cores of the samples were covered with amorphous barium titanate precursors shell (Figure 7A,C,D). Then, the samples were collected by mixing the TEG solution with ethyl acetate and by centrifugation. This solvent is hydrophobic but miscible with polyols and eases the aggregation of nanoparticles. Next, the collected precipitates were dried and subsequently annealed at 700 °C for 5 h to additionally crystallize the obtained composites and increase the number of interfacial connections between magnetic and ferroelectric phases.

The samples after the annealing were studied by XRD and using magnetic measurements. The XRD study revealed that the composites are highly crystalline and consist of barium titanate (PDF Card No.: 01-075-1606), ferrite phase, and barium carbonate (PDF Card No.: 01-078-4343) phases (Figure 8, Table 1). The last phase exists due to the usage of an excess of barium precursor. It should be noted that the average crystallite size of the magnetic phase for x = 0.5-BT and x = 1.0-BT increased from 9.1 ± 0.9 nm and 7.4 ± 0.7 nm to 13.7 ± 1.4 nm and 9.0 ± 0.9 nm, respectively. This happened because of magnetic core recrystallization during annealing. It did not take place for the x = 0.0-BT, probably because of the smaller initial mismatch between crystallite size and the size measured by TEM analysis. All barium titanates are crystallized in the tetragonal phase—the phase, which has prominent ferroelectric properties—with crystallite sizes of 10.1 ± 1.0 nm, 13.9 ± 1.4 nm, and 14.5 ± 1.5 nm for x = 0.0-BT, x = 0.5-BT and x = 1.0-BT, respectively.

The magnetic measurements revealed a significant decrease in the σ_s_ and σ_r_ due to the presence of the non-magnetic phase: barium titanate (Figure 8B, Table 2). The σ_s_ are 13.2 ± 0.1 A*m^2^/kg, 20.2 ± 0.2 A*m^2^/kg and 21.9 ± 0.2 A*m^2^/kg, and the σ_r_ 1.01 ± 0.01 A*m^2^/kg, 4.32 ± 0.04 A*m^2^/kg and 2.24 ± 0.02 A*m^2^/kg for x = 0.0-BT, x = 0.5-BT and x = 1.0-BT, respectively. The σ_s_ decreased not only in absolute values but also in relative values per magnetic phase, especially for the x = 0.5-BT and x = 1.0-BT to 40.8 ± 0.5 A*m^2^/kg and 49.8 ± 0.6 A*m^2^/kg, respectively. This is likely a result of the presence of interfacial defects between ferroelectric and ferrimagnetic phases, that reduce the saturation magnetization [50]. After annealing, the H_c_ decreased slightly to 36 ± 2.5 Oe for the x = 0.0-BT, most likely, due to oxidation of the magnetic core from magnetite to maghemite. While for the x = 0.5-BT and x = 1.0-BT it increased to 240 ± 11 Oe and 89 ± 5 Oe, respectively, because the crystallite size increased after the annealing, which corresponds to the published data [51]. As is well-known, the coercive force of magnetic nanoparticles increases with the increase in average crystallite size above the superparamagnetic limit.

The dependence of ME voltage coefficients ($\alpha_{ME}$) versus bias DC magnetic field $\left(H_{DC}\right)$ for all samples are shown in Figure 9A. As seen all samples demonstrate a peak-like behavior with a maximum at ~1 kOe for samples with x = 0 and ~3.5 kOe for x = 0.5 and 1. The maximums of ME voltage coefficients for samples x = 0.0-BT, x = 0.5-BT, and x = 1.0-BT reached the values; 50, 89, and 74 mV/cm*Oe, respectively. The non-linear dependencies of ME coefficients are associated with magnetostriction and magnetization processes of the magnetic component of composites. As known, the dominant mechanism of ME coupling in MENs is connected to the «magnetostriction *piezoelectric effect» model, when the charge on the surface of the piezoelectric shell results from the mechanical force action, an effect of magnetostriction of the magnetic core when a magnetic field is applied. Therefore, in this case, ME response depends on not only magnetostrictive λ_ij_ and piezoelectric *d_ij_* but also on the rate of mechanical coupling between both components. The significant enhancement of the ME effect in nanocomposites of CoFe_2_O_4_-BaTiO_3_ with core-shell structure up to 8.13 mV/cm*Oe in comparison with mixed composites containing the same components and weight ratio were demonstrated [18]. Similar results were obtained by Duong [52], where, as a result of better interfacial coupling, larger values of ME coefficient ~3.5 mV/cm*Oe in CoFe_2_O_4_-BaTiO_3_ composite with the core-shell structure were observed than in the sample with a mixed structure ~0.2 mV/cm*Oe, and by Rao [9] with 9.18 mV/cm*Oe and 2.84 mV/cm*Oe, respectively. In our case, the samples with x = 0 correspond to the composition with Fe_3_O_4_ core and have a smaller magnetostriction coefficient |λ| ~20 ppm in comparison with CoFe_2_O_4_ based |λ| ~100–200 ppm [53,54]. The samples with x = 0.5 and 1.0 correspond to the compositions with a CoFe_2_O_4_-based core, that is why they demonstrate larger ME coefficients. Additionally, the large ME response is related to good mechanical coupling between CoFe_2_O_4_ and BaTiO_3,_ their optimal ratio, and provides less electrical losses. Moreover, the ferroelectric shell of BaTiO_3_ with a large piezoelectric constant d_33_ = 191 pC/N was reported by J. Gao et al. makes a contribution toward the enhancement of ME coupling in core-shell nanocomposites [55]. It should be noted that the dynamic method used for ME characterization is most suitable for bulk multiferroics and does not present information about the ME response locally induced in core-shell nanograins as a result of the interaction between magnetostrictive core and piezoelectric shell particles. Moreover, the charge leakage problem is typical for 0–3 particulate composites, which is affected by the values of the ME effect. In our studies, the ME measurements were used for preliminary evaluation of ME coupling and comparision with the change in the type of magnetic core Co_x_Fe_3−x_O_4_. To overcome these problems, recently several techniques based on scanning probes have been developed [56,57]. For example, using a technique based on a modified scanning tunneling microscope (STM) probe, a colossal ME coefficient above 5 V cm^−1^ Oe^−1^ in 20 nm CoFe_2_O_4_-BaTiO_3_ core-shell ME nanoparticles was observed, which is larger than in classical 0–3 type bulk composites with the same composition 0.1 V cm^−1^ Oe^−1^ [57].

Despite the lower ME effect for the x = 0.0-BT, the results are significant, meaning that even non-cobalt ferrite-based MENC can be used for biomedical applications, at the same time the iron oxide materials are known to be less toxic and are actively used in biomedicine [58], while usage of cobalt-based materials, being more toxic, is seriously restricted [59,60].

After the annealing, a planetary ball mill was used for dispersing the composites back onto a nanoscale. To ease the process and to get colloidally stable suspensions the samples were milled in a liquid organic mixture of octadecene-1 and oleic acid. Octadecene-1 was chosen because it is a nonpolar alkene with a high boiling point (T_b_ = 320 °C), that helps to avoid side evaporation during ball milling and/or excludes ignition, which is possible in the case of more commonly used flammable liquids. Oleic acid has a very high boiling point as well, but moreover, it can adsorb onto nanocomposite surfaces and plays the role of surface stabilizer, seeing as it is used for ferrite nanoparticle stabilization. To find the optimal time of ball milling the dynamics of size and polydispersity index (PdI) vs. time were studied using the DLS method (Figure 9B). The average size and PdI decreased from 330 nm and 0.72 for one hour of ball milling to 120 nm and 0.14 for 12 h, respectively. These values make the particles suitable for further biomedical testing, because nanostructures greater than 100–200 nm usually show low colloidal stability and can, if used in in vivo experiments, cause an immune response. Then, the samples, after ball milling, were washed out of organic media by mixing them with buthanol-1, twice washing them with isopropanol, and redispersing them in toluene under intensive ultrasonication.

The ball-milled samples were further studied by TEM (Figure 7B,D,F). All samples turned into high crystalline polygranular quasi-spherical structures with a diameter of approximately 100 nm. As standard TEM cannot study the fine structure of MENC, high-resolution TEM was used (Figure 10). In a high-resolution image (Figure 10A) and using electron diffraction (Figure 10B) of x = 0.0-BT the fine structure of the nanocomposite particles is visible: the iron oxide particle is surrounded with several barium titanate particles and both phases have dense interfacial contact, especially, the (113) plane of iron oxide and the (101) plane of barium titanate. An analogous structure of MENC is observed for x = 0.5-BT (Figure 10C,D) and x = 1.0-BT (Figure 10E,F). The ferroelectric and magnetic phases coexist and have interphase connections, which are necessary for a strong ME effect.

### 3.4. Study of the MENC Cytotoxicity on CT-26 Cell Line

For the cytotoxicity studies, the samples were preliminarily transferred from organosol to hydrosol using Pluronic F127 nonionic surfactant. The organosol was diluted to 2 mg/mL by composite and the nanoparticles were hydrophilized via intensive ultrasonication with an aqueous solution of nonionic surfactant Pluronic F127. Then they underwent subsequent centrifugation and the precipitate was redispersed in PBS. The surfactant, as a triblock-copolymer of 2 hydrophilic polyethylene glycol, blocks at its terminals, and hydrophobic polypropylene glycol blocks at the center of the polymer chain adsorb onto the hydrophobic surface of the nanoparticles, which are covered with oleic acid tails, hydrophilize the nanoparticle surface and transfer the nanoparticles into the water media. This polymer is nontoxic and widely used for drug delivery [61,62].

The final three samples of x = 0.0-BT, x = 0.5-BT, and x = 1.0-BT in PBS were tested for cytotoxicity on the CT-26 cancer cell line (Figure 11). The study was carried out in the range of: 25.0–400.0 μg/mL, and a slight increase in cytotoxicity, with a higher amount of Co in MENC, is observed; but the overall decrease in cell viability is small, despite the relatively high toxicity of Ba^2+^ [31] and Co^2+^ ions [29,30,59,60]. This may result from the magnetic phase being covered with barium titanate, which is insoluble in the water of wide pH ranges and persistent toward biological media. Moreover, the extra dense layer of hydrophobic oleic acid and surfactant prevents leakage of the toxic ions into the cell media. 

The cytotoxicity results are comparable with the published data for ME materials of the analogous structure. The synthesized MENC shows lower cytotoxicity on CT-26 than the analogous ME nanoparticles on SKOV-3 [17] Hep2 [63], HepG-2 [20], microglia cells [64], or relatively the same as on astrocytes [65], SKMNC [65], and CHO cells [9]. There have been no cytotoxicity studies of ME nanoparticles performed on CT-26 cell lines yet, therefore there is no more correct data to compare to. Despite this fact, the obtained results show that the synthesized MENC are comparably cytotoxic with the aforementioned analogs or even less so.

## 4. Conclusions

The aim of this research was to study the influence of magnetic core composition on the magnetic properties, the magnetoelectric effect, and the cytotoxicity of magnetoelectric nanocomposites composed of Co_x_Fe_3−x_O_4_-BaTiO_3_, where x = 0; 0.5; 1. This new solvothermal method of ME nanoparticle synthesis has been developed for this reason. The presence of interfacial contacts between magnetic and ferroelectric phases has been confirmed by high-resolution transmission microscopy studies. According to X-ray diffraction, all samples have the ferrite magnetic phase, whereas the nanocomposites also have the BaTiO_3_ phase. According to the vibrating sample magnetometry, all samples are ferrimagnets. It has been found that the coercive force of cobalt-containing nanocomposites increased after annealing. Due to the appearance of a nonmagnetic phase, the saturation magnetization for all samples decreased after annealing. The effect of the various magnetic phases of the nanocomposite on the interaction between the magnetic and ferroelectric phases of the samples has been studied. As a result, the nonlinear influence of the cobalt content on the magnetoelectric properties of the material was revealed, with the increase of cobalt in the magnetic phase, the magnetoelectric coefficient also increases, which is associated with a large value of Co magnetostriction. The highest magnetoelectric coefficient value (89 mV/cm·Oe) was recorded in the Co_0.5_Fe_2.5_O_4_-BaTiO_3_ sample, which combines the higher crystallite size and medium cobalt content. All investigated samples demonstrated high ME properties in a diapason of 50–89 mV/cm·Oe. The cytotoxic studies on the cancer cell line CT-26 revealed that all samples are non-toxic in a concentration range of 25–400 μg/mL. The obtained MENC shows low cytotoxicity and high magnetoelectric effect and can find a wide application in neural stimulation, cell regeneration, hyperthermia, drug delivery, and other biomedical applications.

## Figures and Tables

**Figure 1 nanomaterials-13-00811-f001:**
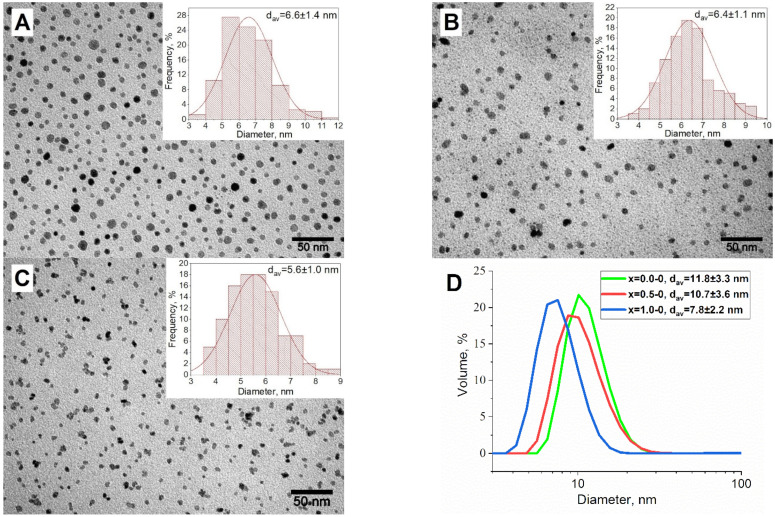
TEM-images and distribution by size for the seed MNP of x = 0.0-0 (**A**); x = 0.5-0 (**B**); x = 1.0-0 (**C**) and their hydrodynamic sizes measurements (**D**).

**Figure 2 nanomaterials-13-00811-f002:**
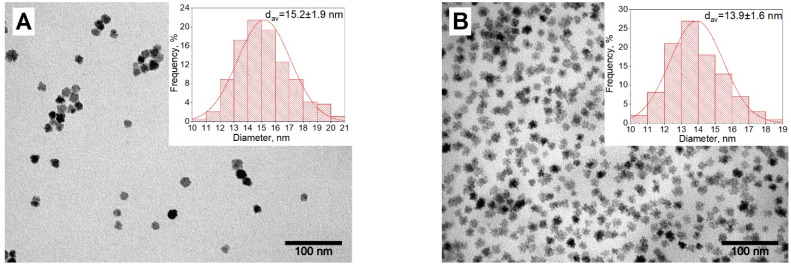
TEM-images and distribution by size for the MNP of x = 0.0-1 (**A**); x = 0.5-1 (**B**); x = 1.0-1 (**C**) and their hydrodynamic sizes measurements (**D**).

**Figure 3 nanomaterials-13-00811-f003:**
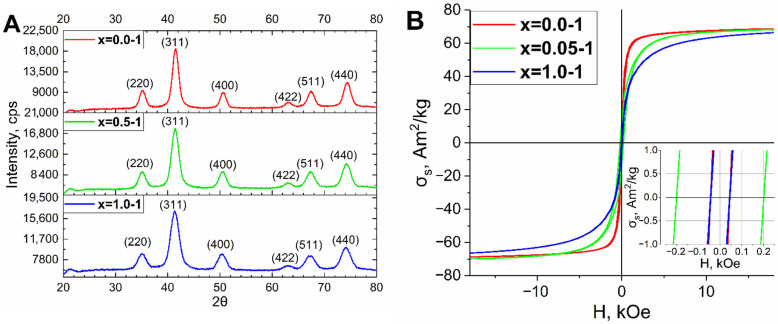
XRD data (**A**) and magnetic properties (**B**) for the MNP of x = 0.0-1; x = 0.5-1 and x = 1.0-1.

**Figure 4 nanomaterials-13-00811-f004:**
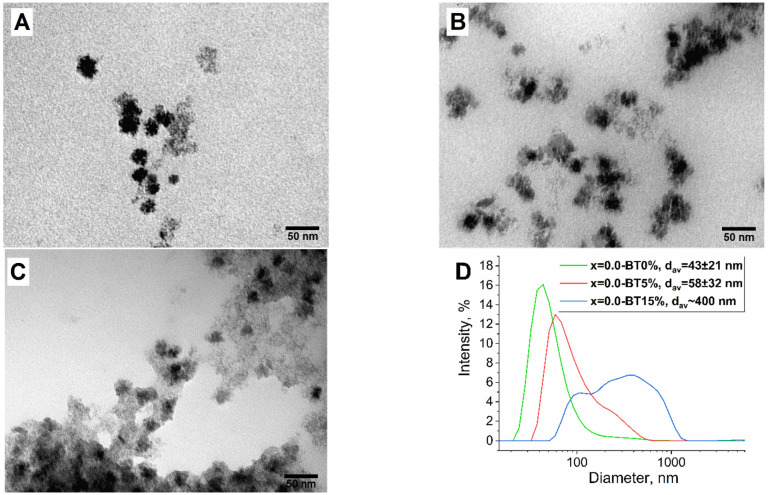
TEM and DLS data for the samples of x = 0.0-BT0% (**A**), x = 0.0-BT5% (**B**), x = 0.0-BT15% (**C**), and their hydrodynamic size (**D**).

**Figure 5 nanomaterials-13-00811-f005:**
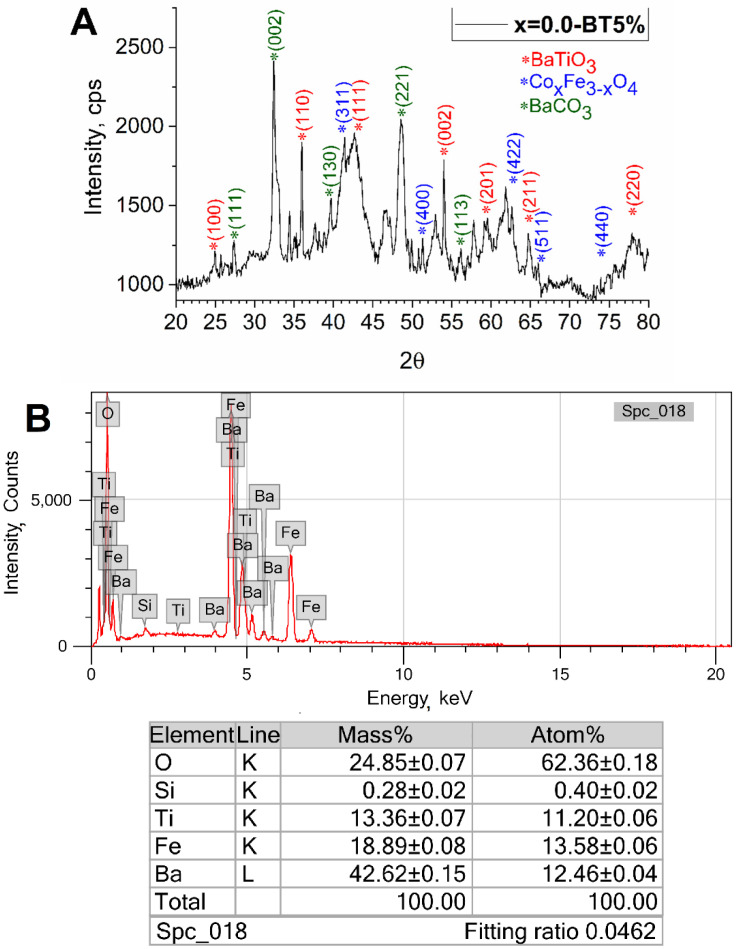
XRD (**A**) and EDX (**B**) study of the obtained x = 0.0-BT5%.

**Figure 6 nanomaterials-13-00811-f006:**
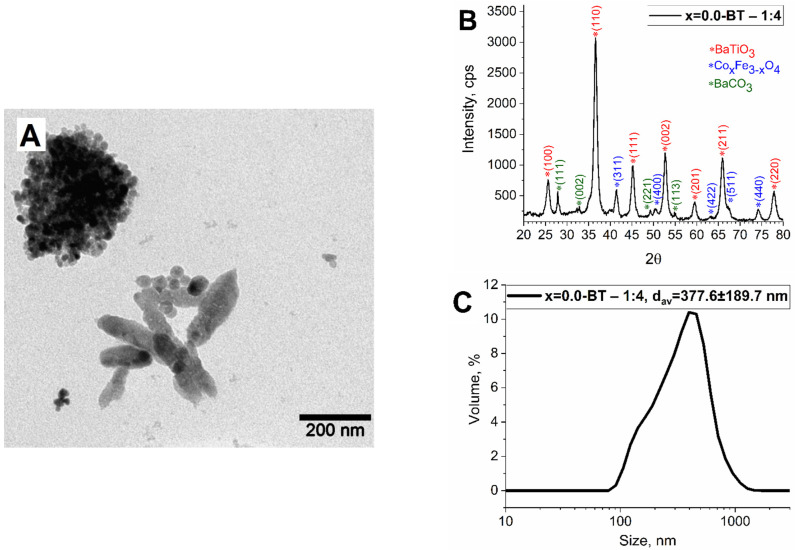
TEM (**A**), XRD (**B**), and DLS (**C**) data for sample x = 0.0-BT—1:4.

**Figure 7 nanomaterials-13-00811-f007:**
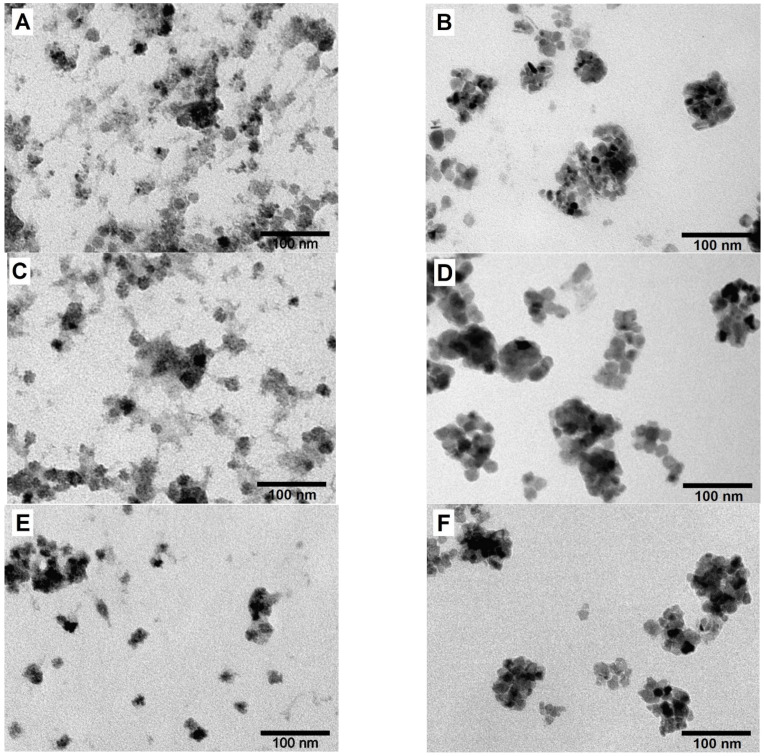
TEM images for x = 0.0-BT, x = 0.5-BT, and x = 1.0-BT samples after hydrothermal synthesis (**A**,**C**,**E**, respectively) and after annealing and ball milling (**B**,**D**,**F**, respectively).

**Figure 8 nanomaterials-13-00811-f008:**
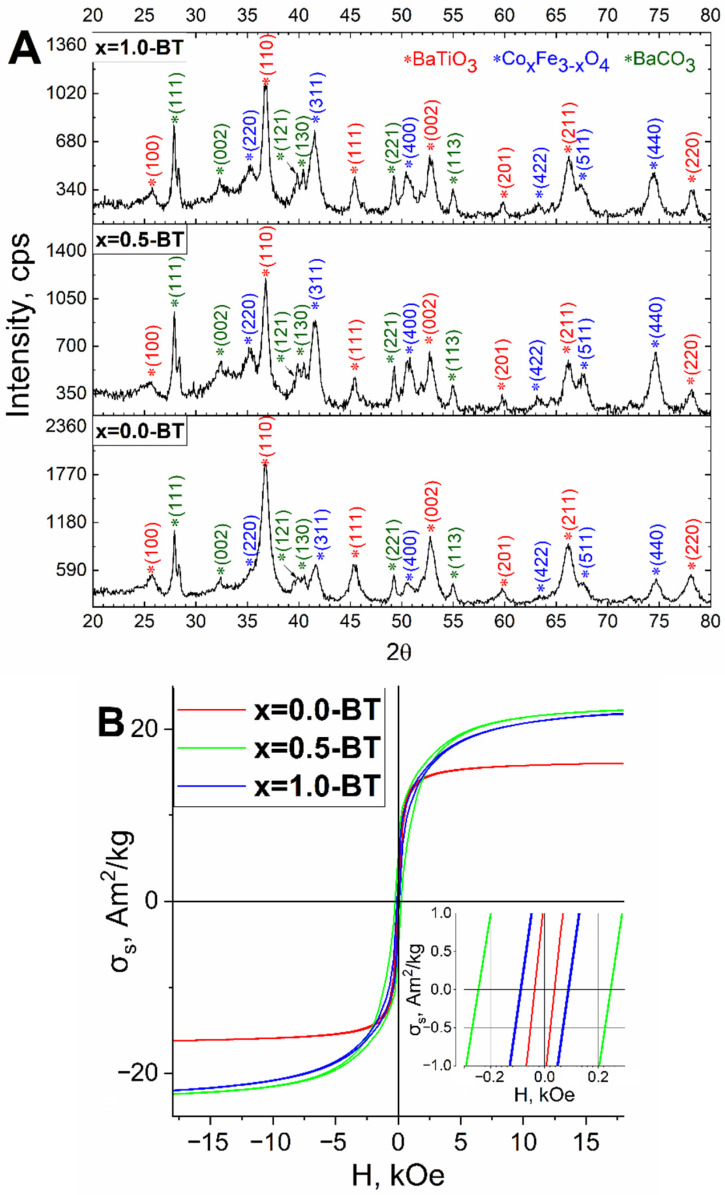
XRD (**A**) and magnetic measurement (**B**) data for the samples of x = 0.0-BT, x = 0.5-BT, and x = 1.0-BT after annealing.

**Figure 9 nanomaterials-13-00811-f009:**
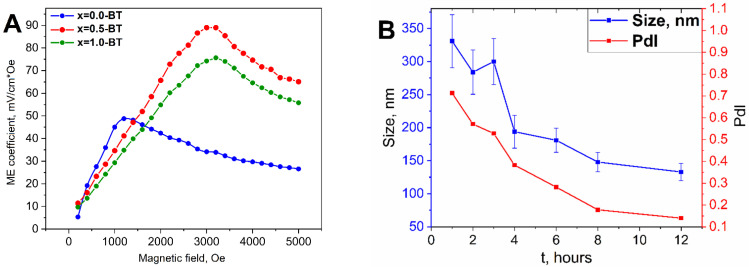
Magnetoelectric measurements (**A**) for the samples of x = 0.0-BT, x = 0.5-BT, and x = 1.0-BT and dynamics of average hydrodynamic size, and PdI change over ball milling time for the x = 0.0-BT by DLS (**B**).

**Figure 10 nanomaterials-13-00811-f010:**
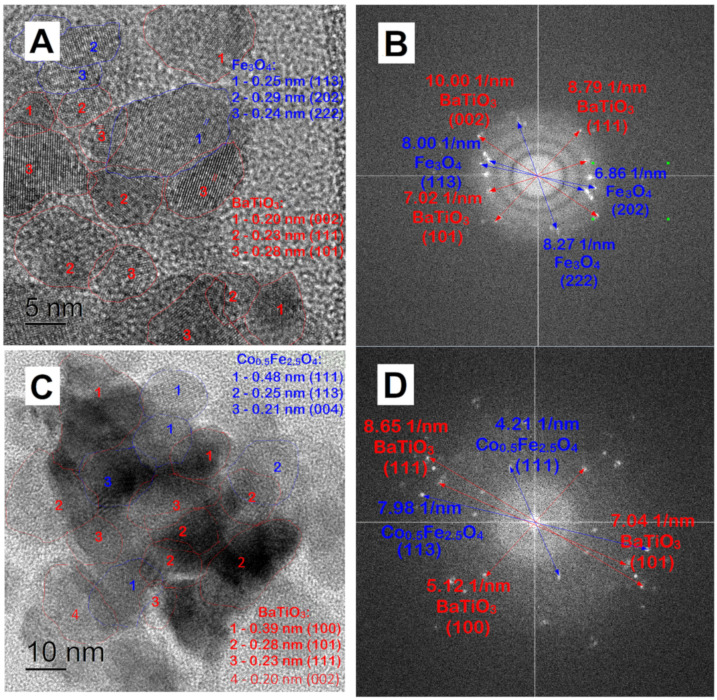
High-resolution TEM images (**A**,**C**,**E**) and electron diffraction (**B**,**D**,**F**) of x = 0.0-BT, x = 0.5-BT, and x = 1.0-BT after ball milling, respectively.

**Figure 11 nanomaterials-13-00811-f011:**
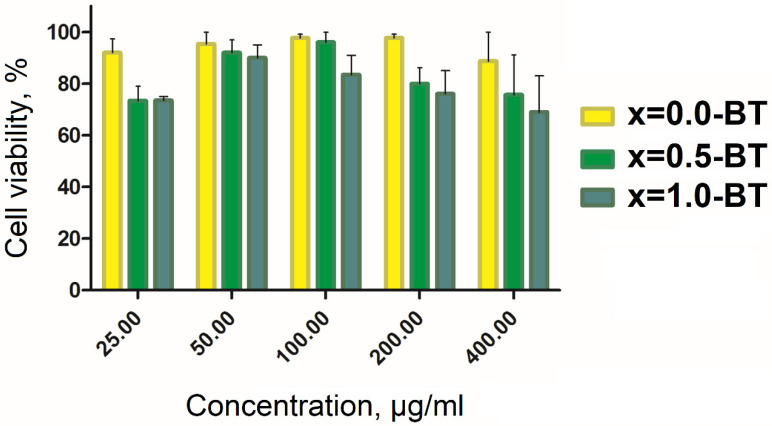
Cytotoxicity study for the samples of x = 0.0-BT, x = 0.5-BT, and x = 1.0-BT on CT-26 cancer cell line.

**Table 1 nanomaterials-13-00811-t001:** XRD data for the synthesized samples.

Sample	Lattice Parameter, Å		Space Group	Phase Composition, %	Crystallite Size, nm
x = 0.0-1	a	8.385 ± 0.001	Fd3m	100% Co_x_Fe_3−x_O_4_	12.4 ± 1.2
x = 0.5-1	a	8.395 ± 0.001	Fd3m	100% Co_x_Fe_3−x_O_4_	9.1 ± 0.9
x = 1.0-1	a	8.412 ± 0.001	Fd3m	100% Co_x_Fe_3−x_O_4_	7.4 ± 0.7
x = 0.0-BT	a	4.019 ± 0.001	99: P4mm	61 ± 3% BaTiO_3_	10.1 ± 1.0
	c	4.018 ± 0.001			
	a	8.376 ± 0.001	227: Fd-3m	20 ± 2% Co_x_Fe_3−x_O_4_	12.8 ± 1.3
	a	6.440 ± 0.001	62: Pnma	18 ± 2% BaCO_3_	41.3 ± 4.1
	b	5.309 ± 0.001			
	c	8.905 ± 0.001			
x = 0.5-BT	a	4.018 ± 0.001	99: P4mm	29 ± 2% BaTiO_3_	13.9 ± 1.4
	c	4.019 ± 0.001			
	a	8.338 ± 0.001	227: Fd-3m	49 ± 3% Co_x_Fe_3−x_O_4_	13.7 ± 1.4
	a	6.439 ± 0.001	62: Pnma	21 ± 2% BaCO_3_	50.4 ± 5.0
	b	5.309 ± 0.001			
	c	8.904 ± 0.001			
x = 1.0-BT	a	4.019 ± 0.001	99: P4mm	43 ± 2% BaTiO_3_	14.5 ± 1.5
	c	4.020 ± 0.001			
	a	8.376 ± 0.001	227: Fd-3m	44 ± 2% Co_x_Fe_3−x_O_4_	9.0 ± 0.9
	a	5.580 ± 0.001	62: Pnma	13 ± 1% BaCO_3_	39.3 ± 3.9
	b	8.902 ± 0.001			
	c	6.431 ± 0.001			

**Table 2 nanomaterials-13-00811-t002:** VSM data for the synthesized samples.

Sample	Phase Composition	H_c_, (Oe)	σ_r_, A•m^2^/kg	σ_s_, A•m^2^/kg	σ_s_ (per Magnetic Phase), A•m^2^/kg
x = 0.0-1	Co_x_Fe_3−x_O_4_ (100%)	49 ± 2.5	5.14 ± 0.05	70.9 ± 0.7	70.9 ± 0.7
x = 0.5-1	Co_x_Fe_3−x_O_4_ (100%)	200 ± 10	7.66 ± 0.08	70.5 ± 0.7	70.5 ± 0.7
x = 1.0-1	Co_x_Fe_3−x_O_4_ (100%)	49 ± 2.5	6.65 ± 0.08	66.2 ± 0.7	66.2 ± 0.7
x = 0.0-BT	BaTiO_3_ (61 ± 3%), Co_x_Fe_3−x_O_4_ (20 ± 2%), BaCO_3_ (18 ± 2%)	36 ± 2.5	1.01 ± 0.01	13.2 ± 0.1	66.0 ± 0.7
x = 0.5-BT	BaTiO_3_ (29 ± 2%), Co_x_Fe_3−x_O_4_ (49 ± 3%), BaCO_3_ (21 ± 2%)	240 ± 11	4.32 ± 0.04	20.2 ± 0.2	40.8 ± 0.5
x = 1.0-BT	BaTiO_3_ (43 ± 2%) Co_x_Fe_3−x_O_4_ (44 ± 2%) BaCO_3_ (13 ± 1%)	89 ± 5	2.24 ± 0.02	21.9 ± 0.2	49.8 ± 0.6

**Table 3 nanomaterials-13-00811-t003:** MS AES data for the x = 0.0-1, x = 0.5-1 and x = 1.0-1 samples.

Sample	C(Co), mg/mL	C(Fe), mg/ml	Empirical Formula
0.0-1	0.00	2.65 ± 0.05	Fe_3_O_4_
0.5-1	0.52 ± 0.05	3.04 ± 0.05	Co_0.42_Fe_2.58_O_4_
1.0-1	3.05 ± 0.05	6.42 ± 0.05	Co_0.93_Fe_2.07_O_4_

## Data Availability

Data are available on request from the corresponding author.

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
