# Peer review of "Synthesis and Functional Characterization of CoxFe3−xO4-BaTiO3 Magnetoelectric Nanocomposites for Biomedical Applications"

_nanomaterials, 2023, doi:10.3390/nano13050811_

Round 1

Reviewer 1 Report

In this manuscript, Timur et al reported the CoxFe3-xO4-BaTiO3 magnetoelectric nanocomposites for biomedical applications, the paper can be accepted after the following issue were concerned.

1. From the XRD results we notice the materials were successfully synthesized, however, will the internal local defect affect the results?

2. In Fig 5A there is a small peak at 41 degree, what is that?

3. From the TEM images it is hard to distinguish whether the structure were amorphours or lattice effect?

4. The performance of this materials should be compared with others' work.

Reviewer 2 Report

In this study, the CoxFe3-xO4-BaTiO3 magnetoelectric nanocomposites were synthesized via a two-step chemical approach in polyol media, and the materials were characterized by TEM, XRD and MP-AES. The cytotoxic test on cancer cell line CT-26 showed that the CoxFe3-xO4-BaTiO3 magnetoelectric nanocomposites are non-toxic in a concentration range of 25-400 mcg/ml.

However, there are some issues in this manuscript. In my opinion, the paper can be published possibly only if the following issues are addressed.

1.        In the last paragraph of Introduction, the advantage of the preparation method of magnetic nanoparticles needs to be clarified.

2.        In section 2.2.1, whether the real content of Co and Fe in CoxFe3-xO4-BaTiO3 magnetic nanoparticles measured by MP-AES was close to the theoretical value? Does the x in CoxFe3-xO4-BaTiO3 represent the real content or theoretical value?

3.        In Fig. 3, it is better to supplement the PDF card for easier verification.

4.        The formats of figures should be unified, such as axes.

5.        In Fig. 7, the phenomenon of magnetic cores covered with amorphous barium titanate precursors shell was hardly observed, which could be characterized by the elements mapping.

6.        In this study, the author only tested the cytotoxic of CoxFe3-xO4-BaTiO3 magnetic nanoparticles on cancer cell line CT-26, some specific biomedical applications of materials should also be conducted.

Reviewer 3 Report

The authors synthesised and characterized CoxFe3-xO4 powders, with x=0.0, 0.5 and 1.0, and CFO-BT powders. The magnetic characterization showed a good agreement between the particle size and the magnetic properties, however the ME characterization were performed on green pellets, which normally have no applicative interest. It is opinion of this reviewer that the authors should address this point and the following suggestions for a possible improving of the manuscript.

11)      The applied resonance frequencies for ME measurements should be reported together with the thickness values of the samples.

22)      Lines 202-218, 306-312 and 362-367 can be moved to the introduction paragraph.

33)      Lines 219-223 and 264-282 can be moved to experimental paragraph.

44)      Lines 239-242: The sentence “all samples have inverse spinel structure” is not supported by the presented data. The authors should measure the degree of inversion.

55)      Lines 253-258: It could be useful if the authors compare the obtained magnetic properties of CFO with those reported in literature (see for example: https://doi.org/10.1016/j.matdes.2016.07.050 ; https://doi.org/10.1021/acs.chemmater.5b01034).

Reviewer 4 Report

Submitted work describes a systematic study of core-shell composite

magnetoelectric of CoxFe3-xO4-BaTiO3 particles obtained by relatively new technique. Despite the fact that manuscript contains very elaborate parts (size distribution analysis for TEM data) and XRD analysis, some other techniques would be nice to employ for better understanding of the material. For example, one could apply XPS technique with step by step surface removal to prove the core-shell structure. For magnetic analysis ZFC-FC data would be most welcome.

The title does not reflect the main content. Authors insist of the definition of their materials as “composites” because the novelty comes from the magnetoelectric effect. However, the materials are rather core-shell particles and they may for composites after cells up-take. Unfortunately the biological testing data do not provide neither optical microscopy, nor TEM for estimation of the way of the particles´ accumulation (see examples: Khlusov, I.A.; Zagrebin, L.V.; Shestov, S.S.; Itin, V.I.; Sedoi, V.S.; Feduschak, T.A.; Terekhova, O.G.; Magaeva, A.A.; Naiden, E.P.; Antipov, S.A.; et al. Colony-forming activity of unipotent hemopoietic precursors under the effect of nanosized ferrites in a constant magnetic field in vitro. Bull. Exp. Biol. Med. 2008, 145, 151–157; Kurlyandskaya, G.V.; Novoselova, I.P.; Schupletsova, V.V.; Andrade, R.; Dunec, N.A.; Litvinova, L.S.; Safronov, A.P.; Yurova, K.A.; Kulesh, N.A.; Dzyuman, A.N.; et al. Nanoparticles for magnetic biosensing systems. J. Magn. Magn. Mater. 2017, 431, 249–254, etc.). These methods might not be available for the authors but at least some discussion of the mechanisms of possible up-take or phase difference in this capacity should be included as supposition.  

The introduction is rather short and not convincing. It should be clearly pointed that proposed materials contain elements which are not really considered for biomedical applications. Data for Fe2O3 or Fe3O4 particles and existing approaches must be mentioned for comparison and explaining the advantages and disadvantages of the proposed materials (Spizzo, F.; Sgarbossa, P.; Sieni, E.; Semenzato, A.; Dughiero, F.; Forzan, M.; Bertani, R.; Del Bianco, L. Synthesis of ferrofluids made of iron oxide nanoflowers: Interplay between carrier fluid and magnetic properties. Nanomaterials 2017, 7, 373, Coisson, M.; Barrera, G.; Celegato, F.; Martino, L.; Vinai, F.; Martino, P.; Ferraro, G.; Tiberto, P. Specific absorption rate determination of magnetic nanoparticles through hyperthermia measurements in non-adiabatic conditions. J. Magn. Magn. Mater. 2016, 415, 2–7 etc.). There too many self-citations for the authors and vary narrow representation of the other contributors – there is no need to increase the number of citations too much but they should be reflecting the existing most important points of view.

There are some technical problems with the manuscript. For example, Fig.8 – Hc is usually for coercivity, for the applied field H symbol is better. Fig. 9 B must include error bars or explain the observed large variations. Authors cannot use different systems for magnetic measurements – field can not be given in Oe for some cases and in kA/m in the others In some graphs/tables the significant numbers are given in the right way, in some they are not correct – careful corrections are necessary.

Proposed composites might be very interesting for different kind applications (still point to proof) but certainly, the future prospects might be described as a broader analysis including magnetic detection with magnetic field sensors, hyperthermia, drug delivery and their combinations.

Reviewer 5 Report

The authors present interesting results using CoxFe3-xO4-BaTiO3 magnetoelectric nanocomposites for biomedical applications. Overall, the manuscript is well written with several outcomes. I would like to recommend for minor revision after I get back from the authors with some of the answers. Here are the comments to the authors.

1. There are several drawbacks and concerns related to nanocomposites for biomedical applications (example. It is difficult to maintain efficiency in the targeted organ once magnetic field is removed). Another disadvantage is the difficulty in maintaining the therapeutic action in three dimensions inside the human body. Despite of the results presented in the manuscript, how the authors ensure the readers about these safety concerns?

2. Please add figure of merit/or literature review.

3. State other biomedical applications of the proposed material.

Round 2

Reviewer 2 Report

Accept.

Author Response

We appreciate your time and effort to provide meaningful review to improve our article. The remarks and advices, that you've given, made the article much better than it was before. 

Reviewer 3 Report

The authors have adressed all the raised points. I think that the manuscript could be accepted in the current version. However I suggest some further improvement: 

Response â„–0: avoid to write "core-shell" since there is not evidence of such microstructure.

Point â„–1: Please report the diameter of the pellets. I am sorry for not pointing this out in the first review.

Point â„–5: I am sorry but in the first review I reported a "wrong" DOI. "https://doi.org/10.1016/j.matdes.2016.07.050" instead of "https://doi.org/10.1016/j.matdes.2017.05.062". It was a copy-paste error due to the proximity of the two articles in the database

Reviewer 4 Report

Authors made reasonable improvements. However the difference between excellent reference work and just a good quality work is here. Authors said "Yes, the ZFC-FC method is more informative, but in our case, we decided to use simpler basic magnetic characterization in order to get Hc, σs and σr." They also made no effort to find ga group for XPS collaboration. The problem with the nanomaterials is that they request many techniques for material undestanding and it is impossible just make a desicion, the nature of material request complete structural and magnetic characterization. Even so, work can be accepted, it is intresting and original, however, could be a top level reseach if better developed. 

Author Response

We are greatful for the remarks and advices, that made the articles much better than it was before. Unfortunately it was difficult to fix all the issues, that we were said about, but we will keep in mind it in the future.